# Data Mining to Atmospheric Corrosion Process Based on Evidence Fusion

**DOI:** 10.3390/ma14226954

**Published:** 2021-11-17

**Authors:** Jintao Meng, Hao Zhang, Xue Wang, Yue Zhao

**Affiliations:** 1Science and Technology on Communication Security Laboratory, Chengdu 610041, China; z.zhhao93@gmail.com (H.Z.); wangxuemath@gmail.com (X.W.); yuezhao@foxmail.com (Y.Z.); 2School of Automation and Electrical Engineering, University of Science and Technology Beijing, Beijing 100083, China

**Keywords:** atmospheric corrosion, carbon steel, data mining, environmental factor, evidence theory

## Abstract

An electrical resistance sensor-based atmospheric corrosion monitor was employed to study the carbon steel corrosion in outdoor atmospheric environments by recording dynamic corrosion data in real-time. Data mining of collected data contributes to uncovering the underlying mechanism of atmospheric corrosion. In this study, it was found that most statistical correlation coefficients do not adapt to outdoor coupled corrosion data. In order to deal with online coupled data, a new machine learning model is proposed from the viewpoint of information fusion. It aims to quantify the contribution of different environmental factors to atmospheric corrosion in different exposure periods. Compared to the commonly used machine learning models of artificial neural networks and support vector machines in the corrosion research field, the experimental results demonstrated the efficiency and superiority of the proposed model on online corrosion data in terms of measuring the importance of atmospheric factors and corrosion prediction accuracy.

## 1. Introduction

Atmospheric corrosion, which may cause enormous economic and human life losses, has occurred in many industries such as infrastructure, energy, and transportation [1,2,3]. Nowadays, variant atmospheric corrosion monitor (ACM) sensors are developed to monitor the real-time corrosion behavior of materials in outdoor dynamic environments [4,5]. The collected big corrosion data contribute to further corrosion science research.

It is generally believed that the mutual behaviors of key environmental factors, including temperature (T), relative humidity (RH), and atmospheric contaminants, lead to atmospheric corrosion [6,7,8]. In order to better understand the atmospheric corrosion process, the real-time variations in the corrosion rate and environmental factors with time should be analyzed. There exist some related works, as summarized in Table 1. Some studies have been carried out in indoor atmosphere environments. By using ACM sensors, the influence of some important factors on the corrosion rate have been verified, such as wet–dry cycles [9] and short-chain volatile carboxylic acids [4]. Considering the complexity of natural atmosphere environments, indoor corrosion tests are now gradually being extended to outdoor environments. Some empirical relationships between corrosion rates and some atmospheric factors have been found out. The long-term atmospheric corrosion of weathering steel bridges has been monitored by an electrochemical impedance-based ACM, finding that the corrosion rate increases gradually with an increasing RH under 80% [10]. The function between the daily charge of ACM sensors and the corrosion rate has been obtained, identifying that the mutual behaviors of driving history, T, and RH cause the corrosion of automobile parts [11]. In another automation corrosion test, the role of salt spray in metal corrosion was confirmed [12]. The obtained data suggested that 75% of zinc corroded in the salt spray phase [12]. The atmospheric corrosion of carbon steel has been monitored by ACM, and the impacts of relative humidity, temperature, and rainfall have been identified to be higher than pollution on initial atmospheric corrosion [13]. Exposure tests with ACM have found that rainfall precipitation is the strongest parameter influencing the initial atmospheric corrosion rate in rainy regions, with RH taking the first place in low-precipitation environments and non-rainfall periods [14]. Recently, some statistical correlation coefficients have been employed to measure the influence of key atmospheric factors in the outdoor atmospheric corrosion process. The maximal information coefficient (MIC) well measures the relationship between some factors (RH, T, and pollutants) and the voltage ratio data [5]. Similarly, Pearson’s correlation coefficient has been employed to identify the effect of important atmospheric parameters [15].

Despite the aforementioned outstanding studies, there still exist some gaps: (1) Concerning the influence of environmental factors on outdoor atmospheric corrosion, most of the existing studies focus on qualitative empirical analysis. Although some statistical correlation coefficients have been used to quantify the influence, unreasonable results may occur. Statistical correlation coefficients can accurately measure the correlation of two variables only when all of the other related variables remain unchanged. However, online ACM corrosion rate data are mutually influenced by many atmospheric factors. Actually, it has been demonstrated that there exist hidden relationships between environmental factors and atmospheric corrosion by a hidden Markov model, which could not be well measured by Pearson’s correlation coefficient [15]. Therefore, how to accurately quantify such underlying relationships is still a challenge. (2) With the development of artificial intelligence, some machine learning models, including artificial neural networks (ANNs) and support vector machines (SVMs), have been widely used in corrosion science [16,17,18,19]. However, these models are not suitable for visually measuring the relationships between key factors and atmospheric corrosion because of their incapable interpretability. They just focus on predicting the corrosion rate by end-to-end model frameworks. Further work to fill the above gaps will contribute to better understanding the atmospheric corrosion mechanism.

Evidence theory [20,21] is a generalization of the traditional probability framework. Given the advantages of information fusion and uncertainty representation, it has been widely used in various tasks [20,21,22]. The main mechanism of evidence theory is to measure and fuse the information provided by multiple evidence. Therefore, evidence theory provides an effective mathematical framework to deal with coupled data because of its information fusion viewpoint.

The corrosion resistance of different materials in the same environment varies greatly, such as carbon steels, alloy steels, and nanostructured materials [23,24]. This study mainly focuses on the real-time atmospheric corrosion of commonly used Q235 steel. Considering that most statistical correlation coefficients do not adapt to outdoor coupled corrosion data, a new evidence fusion-based model is proposed in order to deal with the collected real-time corrosion data of Q235 steel in an outdoor atmospheric environment. It quantifies the different contributions of key environmental factors to the short-term atmospheric corrosion process.

## 2. Mathematical and Experimental Methods

### 2.1. Mathematical Methods

In this subsection, some necessary basics of evidence theory are introduced. Evidence theory, also called Dempster–Shafer theory or belief functions theory, is a generalization of probability framework [20,21]. It is known for the advantages of information fusion and uncertainty representation. Let $H_{q}(q=1,2,\ldots ,c)$ denote c mutually exclusive and exhaustive solutions for an investigated problem. Then $\Omega =\{H_{1},H_{2},\ldots ,H_{c}\}$ is called the frame of discernment for this problem. All of the subsets of Ω compose the power set $2^{\Omega }$. It is important to note that evidence theory is established on the power set. Assume that n pieces of evidence measuring the possible solutions can be obtained. The main framework of evidence theory can be briefly summarized as shown in Figure 1. There exist four steps, and the related definitions are detailed as follows.

Step 1: A basic probability assignment (BPA) function is defined to measure the uncertain information provided by the evidence. The BPA of evidence i is defined as a mapping function $m_{i}$ from $2^{\Omega }$ to the interval [0, 1], such that $m_{i}(\phi )=0$ and:(1)$$\sum_{A\in 2^{\Omega }}m_{i}(A)=1$$

The subset A satisfying $m_{i}(A)>0$ is called a *focal element.* Here, $m_{i}(A)$ measures the degree of support of evidence i to the possible solution A. Moreover, $m_{i}(\Omega )$ indicates the ignorance degree of evidence i to the investigated problem. Obviously, a BPA will degenerate to a probability measure when it only has singleton focal elements.

Step 2: Considering the incomplete reliability of the evidence, a discounting operation is necessary. Let $w_{i}(0\leq w_{i}\leq 1)$ denote the reliability coefficient of evidence i. Then the discounting operation leads to a completely reliable BPA mwii. For $\forall A\in 2^{\Omega }$:(2)miwi(A)={wimi(A)A≠Ωwimi(A)+1−wiA=Ω

Step 3: In order to fuse the multi-source information, different combinations rules have been proposed. Among them, Dempster’s rule [20] is widely used because of its associative and commutative properties. For n discounted BPAs miwi(i=1,2,…,n) provided by n distinct and independent evidence, the fused BPA m=⊕i=1nmiwi satisfies that m(ϕ)=0 and for $\forall A,A_{i}\in 2^{\Omega }$:(3)m(A)=11−K∑∩i=1nAi=A∏i=1nmwii(Ai)
where K=∑∩i=1nAi=ϕ∏i=1nmiwi(Ai) measures the degree of conflict among n discounted pieces of evidence and takes a value in the interval [0, 1]. Dempster’s rule proportionally redistributes the conflicts among all focal elements when K≠1. If K=1, it should be noted that Dempster’s rule will be out of operation and another combination rule could be used.

Step 4: After evidence fusion, the most possible solution for the investigated problem can be determined based on the Pignistic probability BetP related to the BPA m. In order to obtain a solution without ambiguity, this paper just considers $BetP=[BetP_{1},BetP_{2},\ldots ,BetP_{c}]$. For $H_{i}(i=1,2,\ldots ,c)$:(4)$$BetP(H_{i})=\sum_{H_{i}\in B,B\in 2^{\Omega }}\frac{m(B)}{\left|B\right|}$$
where |B| is the cardinality of subset B and $\sum_{i=1}^{c}BetP(H_{i})=1$.

### 2.2. Field Exposure Test

The atmospheric corrosion monitoring tests in this study were conducted at Qingdao (36.07 degrees north latitude and 120.44 degrees east longitude), eastern China, which has a costal humid temperature climate [25]. The principle and availability of the employed Internet of Things Atmospheric Corrosion Monitor (IoTACM) were detailed in our previous research [5]. Following the related instructions [5], the real-time corrosion rate of specimen can be calculated based on the real-time voltage ratio and initial height of the specimen. In this study, three Q235 steel specimens were tested from 25 September 2017 to 24 December 2017, lasting for three months. The sizes (length × width × height) of the A1, A2, and A3 specimens were 70 mm × 70 mm× 0.15 mm, 70 mm × 70 mm × 0.4 mm, and 70 mm× 70 mm × 0.4 mm, respectively. Figure 2 shows some details of the field test. The specimens were installed in an exposure shelf toward the south with a 45-degree angle to the ground. The IoTACM was placed approximately 10 m from the shelf. In addition, humidity and temperature sensors were also placed next to the shelf to record the real-time RH and T of the current environment.

### 2.3. Field Corrosion Data

In this study, the real-time corrosion rates of three Q235 steel specimens and RH and T data were recorded hourly by IoTACM. There were 2200 data samples without missing values. Although only small errors existed among the corrosion rate data of three specimens, the average real-time corrosion rate of A1, A2, and A3 specimens was evaluated so as to improve the reliability of the data. Figure 3a draws the variations of the real-time corrosion rate (*y*-axis) with exposure time. Similarly, Figure 3b draws the variations of cumulative corrosion loss (*y*-axis) with exposure time. Apparently, the corrosion rate significantly fluctuated in a large range and the fluctuation gradually flattened out over time. This is consistent with the power law that the specimens were seriously corroded in the early period, while the corrosion gradually turned slightly with time [26]. In this study, the test period was divided into two stages in order to reveal their underlying different corrosion mechanisms. We took the moment with 50% of the total cumulative corrosion loss as the separation of two stages. Accordingly, as shown in Figure 3a, the corrosion process in test period [0, 550] is defined as Stage 1 (23 days, nearly one month), and the subsequent period [550, 2200] refers to Stage 2. In this case, above 50% of the total cumulative corrosion loss was caused in Stage 1 within only 25% of the total test time. The mean corrosion rate in Stage 1 was 0.021 μm/h, which is 3.5 times that in Stage 2 (0.006 μm/h). It is reasonable to take Stages 1 and 2 as the fast and slow corrosion periods, respectively.

Furthermore, four typical air pollutants—SO_2_, NO_2_, PM2.5, and PM10—and the air quality index (AQI), which are generally regarded as important corrosive agents [5,15], were taken into consideration in this study. The corresponding pollutant data at Qingdao during the same test period were collected from the National Urban Air Quality Real-time Publishing Platform. Moreover the distance between the selected meteorological station and the corrosion test area was 5.4 km. It should be noted that the chloride deposition rate was not included in this work, given the following reasons. First, the impact of chloride is cumulative and becomes more dominant for long-term corrosion under marine atmospheres, while it shows little difference to corrosion losses of carbon steel coupons despite the very different chloride deposition rates in short-term exposure tests [27,28,29]. This study mainly focused on short-term corrosion tests lasting for three months. Second, the average chloride deposition rates in October, November, and December 2017 in the experimental site were 15.38 mgm^−2^d^−1^, 15.91 mgm^−2^d^−1^, and 4.45 mgm^−2^d^−1^, respectively. Compared to the chloride deposition rates ranging from 0∼2000 mgm^−2^d^−1^ reported in related studies [27,28], the above values reflect low chloride concentrations during the exposure tests, which have less effect on the corrosion of Q235 steels in service. Third, the chloride deposition rate is typically recorded at least monthly following ISO 9225 standard [30]. To the best of authors’ knowledge, there is no technique that can precisely record the hourly chloride deposition in field atmospheric environments, especially for our case where the deposition is low and the test duration is short. Last but not the least, the corrosion tests in the same site as this study indicate that the formation of non-protective akageneite induced by chloride enrichment was not found in the corrosion product layer of the specimens after one month of exposure [31]. Additionally, the average chloride deposition rate in reference [31] was 71.03 mgm^−2^d^−1^, which is much higher than the chloride deposition rate during the whole exposure tests in this study. To sum up, it is reasonable to consider that chlorides also did not play a major role in this present study.

Table 2 summarizes the characteristics of the collected field data in each stage. It shows a wide gap in the corrosion rate between Stages 1 and 2. Although the RH in both stages shared similar maximum and standard deviation values, the mean RH in Stage 1 exceeded that in Stage 2. Because of the coming of winter, the T gradually decreased with time. Accordingly, Table 2 shows an obvious lower T in Stage 2 than in Stage 1. According to the data of the contaminators (SO_2_, NO_2_, PM2.5, and PM10), wider variation ranges and bigger average values of pollutants were found in Stage 2. This means that the air pollution in Stage 2 was significantly more serious than in Stage 1.

## 3. Results and Discussion

In this section, the impact of different atmospheric factors on the corrosion rate is analyzed from the viewpoints of qualitative analysis, statistical quantitative analysis, and information fusion-based quantitative analysis.

### 3.1. Qualitative Analysis

In order to visually show the impact of different atmospheric factors on the corrosion rate, the curves of four illustrative cases marked in Figure 3a are drawn. The left *y*-axis in Figure 4 measures the real-time corrosion rate. Meanwhile, the right y-axes in different colors measure the concentrations of corresponding envrionmental factors. As shown in Figure 4b, the decreasing corrosion rate was associated with the gradual reduction in RH and T, while the concentrations of the pollutants remained stable in this period. The period of Figure 4c met with relatively stable values of T and AQI. Moreover, the rise in the corrosion rate with time was significantly influenced by the rapid increase in RH. In the above two cases, the variances of RH and T resulted in thickness variations of the thin electrolyte layer, which directly affected the corrosion rate. Similarly, the RH and T in the periods of Figure 4a,d basically remained stable. However, the air pollutions gradually increased as a whole, especially in the period of Figure 4a. Accordingly, there was a rapid increase in the corresponding real-time corrosion rate. The corrosive components of the pollutants might be dissolved on the surface of specimens, which accelerated the corrosion behavior. The above analyses lead to the conclusion that the RH, T, and these contaminants are positively correlated with the corrosion rate.

### 3.2. Statistical Quantitative Analysis

In order to quantify the influence of atmospheric factors on corrosion rate, the four commonly used statistical correlation indexes of Pearson’s correlation coefficient (PCC) [15], Kendall’s rank correlation coefficient (KRCC) [32], Spearman’s correlation coefficient (SCC) [32], and the maximal information coefficient (MIC) [14] were employed in this subsection. PCC applies to measuring the linear relationship between two variables, SCC and KRCC, focusing on monotonic correlation, while the MIC is capable of measuring nonlinear relationships. The ranges of PCC, SCC, and KRCC are all [−1, 1], where a negative value indicates a negative correlation and vice versa. The MIC takes a value in the [0, 1] range, where a larger value means a stronger correlation between two variables. The results of Stages 1 and 2 are summarized in Table 3 and Table 4, respectively. Given the complexity of corrosion behavior, there exist nonlinear relationships between atmospheric factors and corrosion rate. According to the values of the MIC index in Table 3 and Table 4, the RH and T have stronger correlations with the corrosion rate in both Stages 1 and 2. Compared to Stage 1, the smaller MIC values of the RH and T in Stage 2 indicate their decreased impact. However, the results of the statistical correlation analysis also lead to the following problems:(1)In terms of the influence of the atmospheric factor on the corrosion rate, four statistical indexes derive inconsistent ranks of atmospheric factors in Stage 1. The same thing also occurs in Stage 2. They do not even agree on the most important factor in Stage 2. This indicates that selecting one certain statistical index to evaluate the influences of different atmospheric factors is subjective.(2)In Stage 1, the negative values of T and four contaminators by PCC, KRCC, and SCC indicate their negative correlations with the corrosion rate. However, this is inconsistent with the qualitative analysis results in Section 3.1. Similar things also happen in Stage 2. A possible reason is that statistical correlation coefficients can exactly measure the correlation between two variables only when all of the other related variables remain unchanged. The online corrosion rate data were the interactions of multiple atmospheric variables; therefore, unreasonable statistical results occurred.(3)Regardless of the linear and nonlinear indexes, none of the factors had a correlation value above 0.5. This means low correlations between all factors and the corrosion rate. The possible reason is that the coupled corrosion rate data hinder the statistical indexes from accurately quantifying true correlations. Actually, a hidden relationship between atmospheric factors and the corrosion rate has been demonstrated by a hidden Markov model [15]. How to accurately quantify the hidden influence remains challenging.

### 3.3. Evidence Fusion-Based Quantitative Analysis

From the viewpoint of evidence fusion, a new machine learning model to quantify the hidden influence of atmospheric factors is developed in this subsection. The following model construction process is established in Stage 1, and the results of the proposed model of Stage 2 are directly given.

The corrosion rate data in Stage 1 were classified into the three corrosion degrees of fast, medium, and slow. A commonly used unsupervised K-means method (the function *K Means* in the *Python* package of *Sklearn*) [33] was used to avoid the subjectivity of manual classification. The results are shown in Figure 5. For the real-time corrosion rate $R_{t}$ at exposure time *t*, the corresponding corrosion degree $y_{t}=\left[y^{f},y^{m},y^{s}\right]$ was defined as follows:(5)$$y_{t}=\left\{ \begin{array}{ll} \left[0,0,1\right] & R_{t}\in [0,0.018]\;\\ \left[0,1,0\right] & R_{t}\in (0.018,0.041]\\ \left[1,0,0\right] & R_{t}\in (0.041,0.066] \end{array}\right.$$

Considering that the combined behaviors of multiple factors lead to atmospheric corrosion, the factors of RH, T, SO_2_, NO_2_, PM2.5, and PM10 can be regarded as different independent pieces of evidence providing support information to the current degree of corrosion. It should be noted that AQI is not an independent evidence because it is a synthetic index related to other investigated pollutants. The framework of the proposed model is shown in Figure 6.

Step 1: Evidence construction. Let $x_{i}=[x_{i}^{\mathrm{RH}},\;x_{i}^{T},\;x_{i}^{{\mathrm{SO}}_{2}},\;x_{i}^{{\mathrm{NO}}_{2}},\;x_{i}^{\mathrm{PM}2.5},\;x_{i}^{\mathrm{PM}10}]$ denote the atmospheric factor data sample whose label of the degree of corrosion is $y_{t}=\left[y^{f},\;y^{m},\;y^{s}\right]$. The atmospheric factor dataset on Stage 1 is $X=\{x_{i}{\}}_{i=1}^{550}$. The frame of discernment in this degree of corrosion classification problem is Ω={f, m, s}. Let $X_{train}$ and $X_{test}$ denote the training and test sets, respectively. Let $x^{jk}$ denote the data of the atmospheric factor $j(j\in [\mathrm{RH},\;T,\;{\mathrm{SO}}_{2},\;{\mathrm{NO}}_{2},\;\mathrm{PM}2.5,\;\mathrm{PM}10])$ belonging to the degree of corrosion k(k∈Ω). Moreover, $UB_{jk}$ and $LB_{jk}$ denote the maximum and minimum value of $x^{jk}$ in the training set, respectively. Considering of the universality of Gaussian distribution in the natural world, the Gaussian kernel density estimator [34] was employed to calculate the probability density function (PDF) $f_{jk}$ in the training set:(6)$$f_{jk}(x)=\left\{ \begin{array}{ll} \frac{1}{n_{train}}\sum_{x^{jk}\in X_{train}}\frac{1}{\sqrt{2\pi h}}\exp^{-{\left(x-x^{jk}\right)}^{2}/2h} & \mathrm{if}\;x\in [LB_{jk},\;UB_{jk}]\\ \delta_{0} & \mathrm{otherwise} \end{array}\right.$$
where $h={\left(4\sigma /3n_{train}\right)}^{2}$, σ is the standard deviation of $x^{jk}$ in the training set and $n_{train}$ is the total number of samples in the training set [34]. It should be noted that $\delta_{0}$ is a positive value close to zero so as to avoid generating completely conflicting evidence. In this study, let $\delta_{0}$ = 0.001.

As shown in Figure 6, the support information for the degree of corrosion k provided by evidence j is proportional to its intersection $f_{jk}(x_{i}^{j})$ with the PDF model $f_{jk}$ [35]. Accordingly, the basic probability assignment function $m_{i}^{j}$ of evidence j can be calculated by two rules in reference [35].

Step 2: Evidence discount. Considering the different levels of importance of different pieces of evidence, the evidence discounting operation is necessary to obtain reasonable fusion results. Let $w_{j}(0\leq w_{j}\leq 1)$ denote the importance coefficient of evidence j whose initial value equals 1 before training. For evidence $m_{i}^{j}$, the discounted evidence mwjij can be calculated as follows. For ∀A⊂Ω:(7)mwjij(A)={wj⋅mij(A)if A≠Ωwj⋅mij(A)+1−wjotherwise

Steps 3 and 4: Evidence fusion and corrosion prediction. The different pieces of discounted evidence can be fused by Dempster’s rule [20].
(8)mi=⊕j=16mwjij

Then, the predicted degree of corrosion $BetP_{i}=[BetP_{i}^{f},\;BetP_{i}^{m},\;BetP_{i}^{s}]$ can be derived by the Pignistic probability transformation [36]. For ∀k∈Ω:(9)$$BetP_{i}^{k}=\sum_{k\in B,\;B\subset \Omega }\frac{m_{i}(B)}{\left|B\right|}$$
where |B| is the cardinality of set B.

In order to determine the correct importance $w(w=[w_{1},\;w_{2},\;\ldots ,w_{6}])$ of atmospheric factor evidence, the vector w was optimized on the training set. The objective function $f_{\mathrm{cost}}$ is devoted to minimizing the error between the predicted degree of corrosion and the true degree of corrosion on the training set.
(10)$$f_{\mathrm{cost}}=\underset{w}{\min}\sum_{i=1}^{n_{train}}{\left\|BetP_{i}-y_{i}\right\|}_{2}^{2}$$

By the commonly used simulated annealing optimization method [37], the optimal importance vector $w^{*}$ was derived and used to verify the proposed model on the test set.

In this study, 80% of the data in each stage was randomly selected as the training set and the remainder as the test set. The widely used models of ANN and SVM were also tested. The average prediction precision of these three models on the test set in five random experiments was compared. The importance $w^{*}$ of each atmospheric factor given by our model is reported.

The experimental results of Stage 1 are given in Table 5 and Table 6. The corresponding analyses are as follows: (1) According to Table 5, from the viewpoint of evidence fusion, T is the greatest contributor to atmospheric corrosion in Stage 1, followed by RH and SO_2_. (2) Comparing RH and T, the contaminators of SO_2_, NO_2_, PM10, and PM2.5 have less influence in the interactions on atmospheric corrosion in the initial atmospheric corrosion process. (3) Comparing the statistical correlation coefficients in Table 3, the proposed model found higher correlations between the atmospheric factors and corrosion rate. (4) According to Table 6, comparing ANN and SVM, the proposed model performed best in predicting the degree of corrosion of Q235 steel.

Similarly, the experimental results of Stage 2 are summarized in Table 7 and Table 8. The corresponding analyses are as follows: (1) According to Table 7, in Stage 2, T still contributes most to atmospheric corrosion among all of the investigated factors, followed by NO_2_ and SO_2_. (2) According to Table 2, the mean RH in Stage 2 is obviously lower than in Stage 1. Accordingly, the proposed model derived a lower impact of RH in Stage 2 than in Stage 1. (3) As introduced in Table 2, the test period of Stage 2 suffered more serious air pollution. Accordingly, compared to Stage 1, the proposed model found higher correlations between the corrosion rate and contaminators of NO_2_, PM2.5, and PM10 in Stage 2. (4) According to Table 7, the contaminators of SO_2_ and NO_2_ have more influence on atmospheric corrosion than RH. Following Figure 3b, the corrosion behavior in Stage 2 gradually weakened over time. The possible reason is that Stage 1 generated corrosion products. They isolated the metal surface from the atmosphere such that the adhesion of water droplets on the metal surface was affected. Therefore, the impact of RH was weakened in Stage 2, while some contaminators contributed more to the specimen’s corrosion because of the ability of damaging the existing corrosion products. (5) According to Table 8, the proposed model outperformed SVM and ANN in terms of corrosion predication.

To sum up, according to the results of the qualitative analysis and the statistical quantitative analysis on the exposure test data of Q235 steel at Qingdao, China, it was found that most statistical correlation coefficients did not adapt to the outdoor coupled corrosion data. Therefore, a new evidence fusion-based model was proposed. According to the results of the evidence fusion-based quantitative analysis, the proposed model can discover the influence of different environmental factors on carbon steel corrosion in different exposure test periods, and can accurately predict the corrosion rate.

## 4. Conclusions

Based on the exposure tests of Q235 steel at Qingdao, China, the main findings and novelty of this work are as follows:(1)It was found that most statistical correlation coefficients do not adapt to coupled data analysis. They can accurately measure the correlation between two variables only when all of the other related variables remain unchanged, while the outdoor online ACM corrosion data are mutually influenced by many atmospheric factors.(2)A new evidence fusion-based machine learning model was proposed, which adapts to deal with coupled corrosion data because of the advantage of information fusion. Processing online corrosion data from the viewpoint of evidence theory initiates a new field of corrosion research.(3)The proposed model can not only measure the influence of different environmental factors on atmospheric corrosion, but can also predict the corrosion rate.(4)Comparing the commonly used machine learning models of ANN and SVM in corrosion research field, the proposed model can obtain more accurate corrosion prediction results.(5)According to the proposed model, relative humidity, temperature, and SO_2_ are the main factors affecting atmospheric corrosion among the investigated environmental factors.(6)According to the proposed model, because of the ability of damaging the corrosion products generated in initial corrosion process, SO_2_ and NO_2_ showed greater impact on atmospheric corrosion than relative humidity in the later corrosion period.

Our future work will focus on the atmospheric corrosion analysis on different climatic conditions and long-term corrosion tests.

## Figures and Tables

**Figure 1 materials-14-06954-f001:**
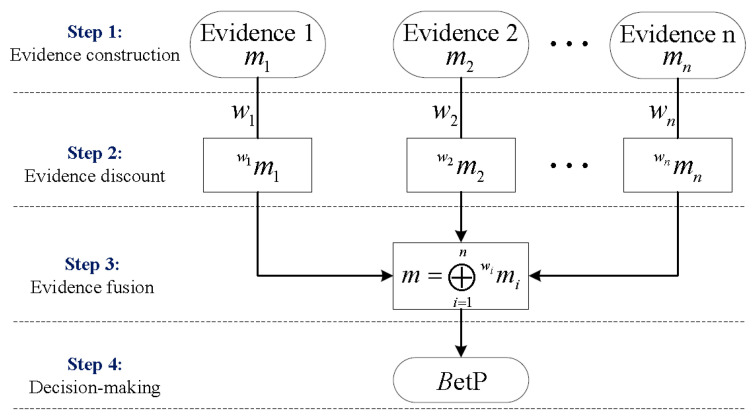
The main framework of evidence theory.

**Figure 2 materials-14-06954-f002:**
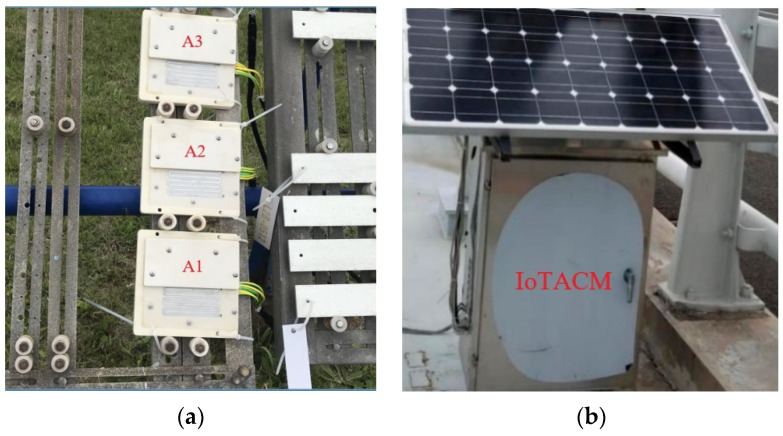
The field corrosion tests at Qingdao, China. (**a**) Three Q235 steel specimens; (**b**) ACM.

**Figure 3 materials-14-06954-f003:**
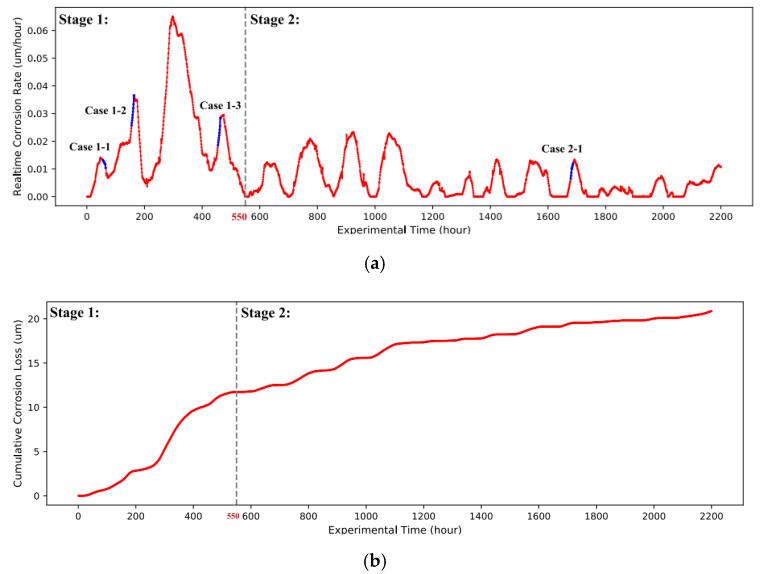
Variations of corrosion data with exposure time. (**a**) Corrosion rate (μm/h); (**b**) cumulative corrosion loss (μm).

**Figure 4 materials-14-06954-f004:**
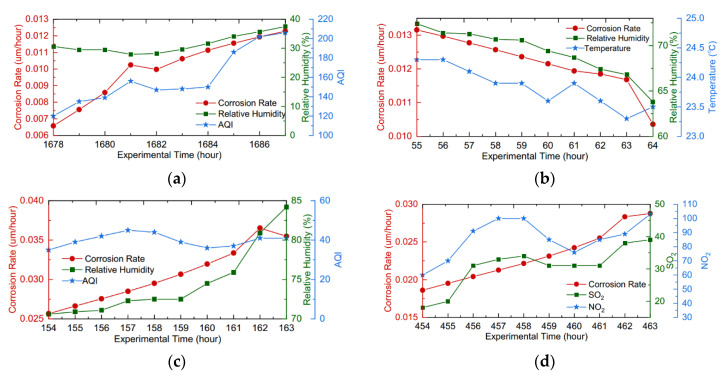
Curves of the four illustrative cases marked in Figure 3a: (**a**) Case 2-1; (**b**) Case 1-1; (**c**) Case 1-2; (**d**) Case 1-3.

**Figure 5 materials-14-06954-f005:**
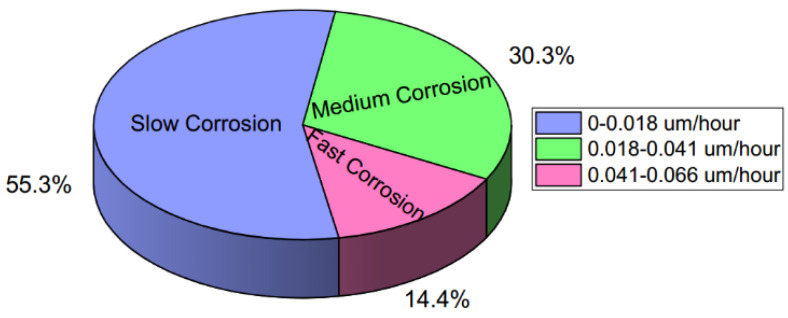
The classification of the corrosion rate on Stage 1.

**Figure 6 materials-14-06954-f006:**
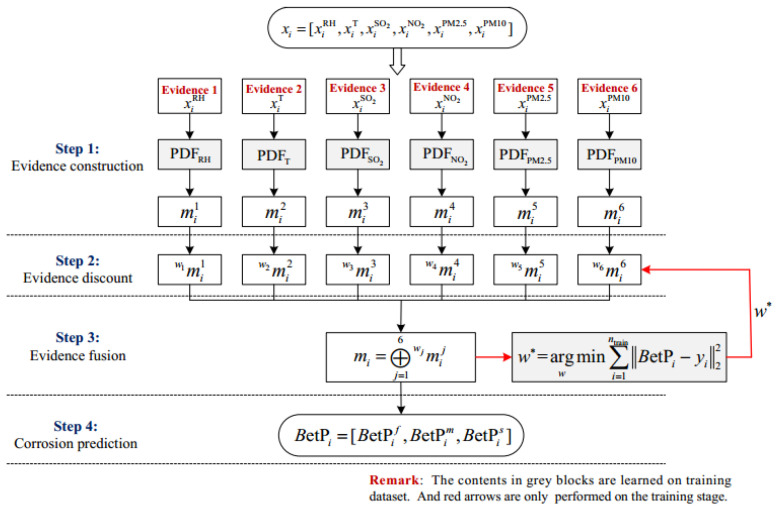
The framework of the proposed model.

**Table 1 materials-14-06954-t001:** Summary of some related works with ACM data.

Papers	Condition	Environment	Atmospheric Factors	Year
[9]	Indoor	Atmosphere	Wet–dry cycles	2014
[4]	Indoor	Atmosphere	T, RH, and formic and acetic acid	2014
[10]	Outdoor	Atmosphere	T, RH	2014
[11]	Outdoor	Automobile	T, RH, and salt particles	2014
[12]	Outdoor	Automobile	T, RH, salt spray, and freeze	2014
[15]	Outdoor	Atmosphere	T, RH, SO_2_, NO_2_, PM2.5, and PM10	2018
[5]	Outdoor	Atmosphere	T, RH, SO_2_, NO_2_, PM2.5, and PM10	2019
[13]	Outdoor	Atmosphere	T, RH, Rainfall, SO_2_, NO_2_, PM2.5, PM10, O_3_, and CO	2020
[14]	Outdoor	Atmosphere	T, RH, and rainfall	2021

**Table 2 materials-14-06954-t002:** Characteristics of the field data in Stage 1 (Stage 2).

Index	R_t_	RH	T	AQI	SO_2_	NO_2_	PM2.5	PM10
Max	0.065 (0.023)	84 (79)	26 (19)	97 (289)	45 (163)	112 (412)	28 (98)	131 (135)
Min	0 (0)	25 (17)	17 (1)	20 (5)	1 (1)	8 (1)	1 (1)	1 (4)
Mean	0.021 (0.006)	58 (42)	21 (10)	44 (76)	13 (37)	45 (97)	6 (15)	29 (51)
Standard Deviation	0.017 (0.006)	16 (14)	2 (5)	14 (32)	10 (31)	21 (51)	5 (11)	22 (28)

Notes: R_t_ is the corrosion rate at time *t*. The units of R_t_, RH, and T are μm/h, *%*, and °C, respectively. The units of SO_2_, NO_2_, PM2.5, and PM10 are all µg/m^3^.

**Table 3 materials-14-06954-t003:** Statistical correlations of atmospheric factors and corrosion rate in Stage 1.

Index	RH	T	AQI	SO_2_	NO_2_	PM2.5	PM10
PCC	0.4765	−0.2801	−0.2347	−0.1932	−0.2584	−0.3026	−0.2150
KRCC	0.3350	−0.1661	−0.1568	−0.1055	−0.1792	−0.2374	−0.1724
SCC	0.4789	−0.2060	−0.2453	−0.1593	−0.2695	−0.3300	−0.2582
MIC	0.4225	0.3373	0.2889	0.3079	0.2685	0.3191	0.2876

**Table 4 materials-14-06954-t004:** Statistical correlations of atmospheric factors and corrosion rate in Stage 2.

Index	RH	T	AQI	SO_2_	NO_2_	PM2.5	PM10
PCC	0.3113	0.2812	0.1188	0.1059	0.1308	−0.0285	−0.2019
KRCC	0.2222	0.1471	0.0937	0.0786	0.0855	−0.0052	−0.1545
SCC	0.3271	0.2265	0.1344	0.1229	0.1220	−0.0067	−0.2250
MIC	0.2108	0.2493	0.1475	0.1661	0.1476	0.1327	0.1827

**Table 5 materials-14-06954-t005:** The importance of each atmospheric factor in Stage 1 (after 550 hours of exposure).

Factor	RH	T	SO_2_	NO_2_	PM2.5	PM10
$w^{*}$	0.9031	0.9881	0.8910	0.4491	0.1175	0.4208

**Table 6 materials-14-06954-t006:** The average prediction precision in Stage 1 (after 550 hours of exposure).

Algorithm	SVM	ANN	Ours
Accuracy	62.00%	75.63%	75.63%

**Table 7 materials-14-06954-t007:** The importance of each atmospheric factor in Stage 2 (after 2200 hours of exposure).

Factor	RH	T	SO_2_	NO_2_	PM2.5	PM10
$w^{*}$	0.7821	0.9601	0.8547	0.9464	0.7146	0.6385

**Table 8 materials-14-06954-t008:** The average prediction precision in Stage 2 (after 2200 hours of exposure).

Algorithm	SVM	ANN	Ours
Accuracy	74.44%	72.57%	80.88%

## Data Availability

The raw/processed data required to reproduce these findings cannot be shared at this time as the data also form part of an ongoing study.

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
