# Peer review of "Data Mining to Atmospheric Corrosion Process Based on Evidence Fusion"

_materials, 2021, doi:10.3390/ma14226954_

Round 1
Reviewer 1 Report
Nice work but it needs minor revision.
- In introduction: Need to update the role of nanostructured materials in corrosion.
- In table 1: The authors should add the recent work related to ACM data.
- The novelty of the work should be established.
- In figures 3 and 4: check the y-axis carefully.
- The authors should include the exposure test graph of PM5 and PM10 concentrations.
- The authors must add the importance index of environmental factors to IACM after one-month exposure.
- The authors should add the prediction results of different periods for the testing samples.
- Improve the conclusion part.
- English language should be improved.
- Typo errors in the manuscript. Check it very carefully.
Reviewer 2 Report
The authors proposed a new machine-learning model to quantify the contribution of different environmental factors to atmospheric corrosion. An electrical resistance sensor-based atmospheric corrosion monitor was employed to study the corrosivity of outdoor atmospheric environments by recording dynamic corrosion data in real-time. Effective and efficient data mining models for the collected data contributed to uncovering the underlying mechanism of atmospheric corrosion. In the study, the factors of relative humidity, temperature and contaminants were regarded as multiple evidence measuring the atmospheric corrosion. The subject matter is important and has a special value considering the practical applications, especially because of developing a new evidence fusion-based machine learning model to quantify the contribution of different environmental factors to atmospheric corrosion. The paper is rather clearly presented and well organized. The tables and figures are adequate. The conclusions seem to be sound and justified. However, I suggest a minor revision regarding the following points to increase the quality of the paper:
- The caption of Figure 4 should be: “Figure 4. Curves of four illustrative cases marked in Figure. 2-(a). (a) Case 2-1; (b) Case 1-1; (c) Case 1-2; (d) Case 1-3.”
- In the scientific work the use of phrases such as “we develop…”, “We propose…” or “we draw…” is not advisable. The impersonal sentences should be used.
Reviewer 3 Report
"Towards data mining to short-term atmospheric corrosion process via machine learning" The Authors present and discuss some results on an electrical resistance sensor-based atmospheric corrosion monitor, employed to study the corrosivity of outdoor atmospheric environments by recording dynamic corrosion data in real-time. Furthermore, the Authors employ data mining models for the collected data, that contribute to uncover the underlying mechanism of atmospheric corrosion. The manuscript is well written, however several aspects need thorough revision. * Title: not meaningful; please rephrase it to make more accurate with respect to your actual work. * Keywords: not meaningful / insufficient; provide more meaningful ones. * Abstract: not meaningful (rather seems to be an introduction, not an abstract); please rephrase it and insist more on the novelty and importance of your approach; avoid using any sort of speculative words and/or phrases (e.g. "Effective and efficient", ...). 1. Introduction * this section is not sufficiently developed; the Authors need to insist more on the novelty and importance of their approach with respect to literature (some important publications are missing): further explain on your approach, and also consider providing further references to it. * the last paragraph is a brief presentation of your work (and should not discuss on the structure of the paper!), insufficiently developed / not enough details are provided. 2. Mathematical and Experimental Methods * please merge and rename sections "2. Preliminaries" and "3. Experiments" as such (recommended). * as a general remark, some of the mathematical and experimental methods in this section need to be further discussed; however, there's no need to present well-known techniques (references will suffice); * include here all mathematical and experimental details, use subsections if needed; provide further references where available, to better understand your approach; avoid redundant text; * Figure 2: provide scale bars; * Figure 3: further discuss this in text; same comment for Table 2. 3. Results and discussion * as a general overview / remark to this section: the Authors discuss their results in a good and well, correlated manner; however, please further present and discuss in text all figures and tables, provide references where available. * a final (last) paragraph of section 3 must be included, to provide the reader with a brief conclusion of your work / manuscript (and insist more on the novelty of your approach). 4. Conclusion * this section is meaningful, yet too short; please rephrase it to emphasize more on the novelty and importance of your approach; provide further data relevant to this study. To conclude, the manuscript should be considered for publication only after careful revision.
Round 2
Reviewer 3 Report
Original title: "Towards data mining to short-term atmospheric corrosion process via machine learning"
Revision 1 title: "Evidence fusion-based data mining to atmospheric corrosion process"
The Authors have correctly addressed most of the issues raised during the peer-review-process. The manuscript is now suitable for publication in MDPI's journal "Materials".